# Correlation via Synthesis: End-to-end Image Generation and Radiogenomic Learning Based on Generative Adversarial Network

**Ziyue Xu**                                    ZIYUEX@NVIDIA.COM

**Xiaosong Wang**                         XIAOSONGW@NVIDIA.COM

**Hoo-Chang Shin**                             HSHIN@NVIDIA.COM

**Dong Yang**                                    DONGY@NVIDIA.COM

**Holger Roth**                                  HROTH@NVIDIA.COM

**Fausto Milletari**              FAUSTO.MILLETARI@GMAIL.COM

**Ling Zhang**                       ZHANGLING0722@163.COM

**Daguang Xu**                               DAGUANGX@NVIDIA.COM

## Abstract

Radiogenomic map linking image features and gene expression profiles has great potential for non-invasively identifying molecular properties of a particular type of disease. Conventionally, such map is produced in three independent steps: 1) gene-clustering to metagenes, 2) image feature extraction, and 3) statistical correlation between metagenes and image features. Each step is separately performed and relies on arbitrary measurements without considering the correlation among each other. In this work, we investigate the potential of an end-to-end method fusing gene code with image features to generate synthetic pathology image and learn radiogenomic map simultaneously. To achieve this goal, we develop a multi-conditional generative adversarial network (GAN) conditioned on both background images and gene expression code, synthesizing the corresponding image. Image and gene features are fused at different scales to ensure both the separation of pathology part and background, as well as the realism and quality of the synthesized image. We tested our method on non-small cell lung cancer (NSCLC) dataset. Results demonstrate that the proposed method produces realistic synthetic images, and provides a promising way to find gene-image relationship in a holistic end-to-end manner.

**Keywords:** Image synthesis, Radiogenomic learning, Multi-conditional GAN

## 1. Introduction

The integration of genomic and imaging findings has emerged as a promising direction in clinical research, often being referred to as radiogenomics. A few studies has been performed to examine its potential in several diseases with different imaging techniques, including magnetic resonance (MR) in brain tumor (Diehn et al., 2008), and computed tomography (CT) in nonsmall cell lung cancer (NSCLC) (Zhou et al., 2018).

Despite the differences in disease and imaging modality, most current studies shared a common methodology in radiogenomics map generation. It is often designed following three independent steps: 1) image feature extraction, either computationally derived (Gevaert et al., 2012), or manually annotated (Zhou et al., 2018); 2) metagene clustering from the gene expression data on the basis of coexpression; and 3) statistical analysis to identify associations between image features and metagenes.

Although it is shown to be a viable way for radiogenomics, it may potentially have some limitations. First, image features are either existing hand-crafted sets or manually defined semantic judgements. The former may not be an optimal representation for the candidate data, and the later can in addition suffer from inter- and intra- observer variability. Second, the metagene clustering is based on statistical correlation analysis, which depends on the specific model being used, and may miss some correlation during model application. Third and most importantly, the image features and genetic characteristics are treated independently without the knowledge of each other, and they can only be correlated in the last step. Hence, sub-optimal image representation, and model-dependent genome clustering may lead to weak correlations that can have limited power in reflecting reality. To out best knowledge, there is no prior work treating all three steps in an alternative way: holistic and end-to-end.

Major challenges in holistically learning a radiogemonic map include: 1) from gene's point of view, it is difficult to fuse the non-image genomic information with image so that the two can be correlated within a single system; 2) from image's of view, it is difficult to use an arbitrary feature extractor to describe its feature related to its corresponding genomic representation, and 3) background region beyond the lesion may be irrelevant to the disease, hence a proper object/background separation is needed; however, the "interaction region" can also hold significant value in lesion characterization, and therefore directly applying a binary segmentation may not be an optimal solution.

With recent development of deep learning, image features can be learnt from data and be optimized for a specific task (Shin et al., 2016). Comparing with hand-crafted or semantic features, such learnt feature presents higher accuracy and robustness in several tasks. Meanwhile, generative adversarial network (GAN) has enabled computer vision researchers to artificially generate realistic images from noise, reference images, and word embeddings (Zhang et al., 2017). A few successes have also been achieved in medical domain (Jin et al., 2018). GAN features the capability of fusing information from different sources to generate the output.

In this work, we propose to use image synthesis as a "bridge" to connect image data with genomic representation, so that we can address some of the challenges. We investigate the potential of a multi-conditional GAN designed for holistically analyze gene expression data and medical images. By utilizing them for new sample generation, both image features and gene embeddings can be learnt directly from data in an end-to-end manner. Also, the design of the network ensures smooth object/background fusion so that only meaningful lesion information get correlated with genomic data. As a proof-of-concept study, we applied our strategy to a public NSCLC dataset with gene expression profiles from RNA sequencing. NSCLC is a common type of lung cancer and leading cause of mortality, and it is known that both imaging and gene expression play important role in its management.

The major contributions of this work are 1) we formulate image-gene correlation by solving a multi-conditional GAN, 2) we design a new GAN architecture and fusion blocks to separate irrelevant image background from lesion so that gene only controls the lesion objects, and 3) we demonstrate that a discriminative radiogemonic map can be learnt via this synthesis strategy.

## 2. Method

In this work, we approach radiogemonic map by formulating it as an image synthesis task. Existing method for CT lung nodule simulation are mostly modeled as "inpainting" based on conditional GAN with no (Jin et al., 2018) (Liu et al., 2018) or limited (Yang et al., 2018) ability in combin-

ing other information beyond the surrounding image. There are two major challenges in applying such network for radiogemonic purpose: 1) there is no direct mechanism to introduce non-image information to the network, which is required for genomic data; and 2) inpainting removes part of the image making space for the simulated nodule, the regional information is lost, and hence it is difficult to ensure spatial continuity and to avoid artificial looking around the boundary of erased region. Therefore, there's no guarantee that the synthesized image have good object/background separation coded within the network, which is critical for radiogenomics learning.

To address these issues, inspired by computer vision works for natural image synthesis (Park et al., 2018) (Karras et al., 2018), we design our network as a multi-conditional GAN with style specification. Foreground/background fusion is modeled within network, while background image and gene coding are both utilized for synthesis. Fig.1 illustrates an overview of our method. Below, we outline the GAN architecture, information fusion design, and training strategy for learning the representation and generating lung nodules from both imaging and genomic data.

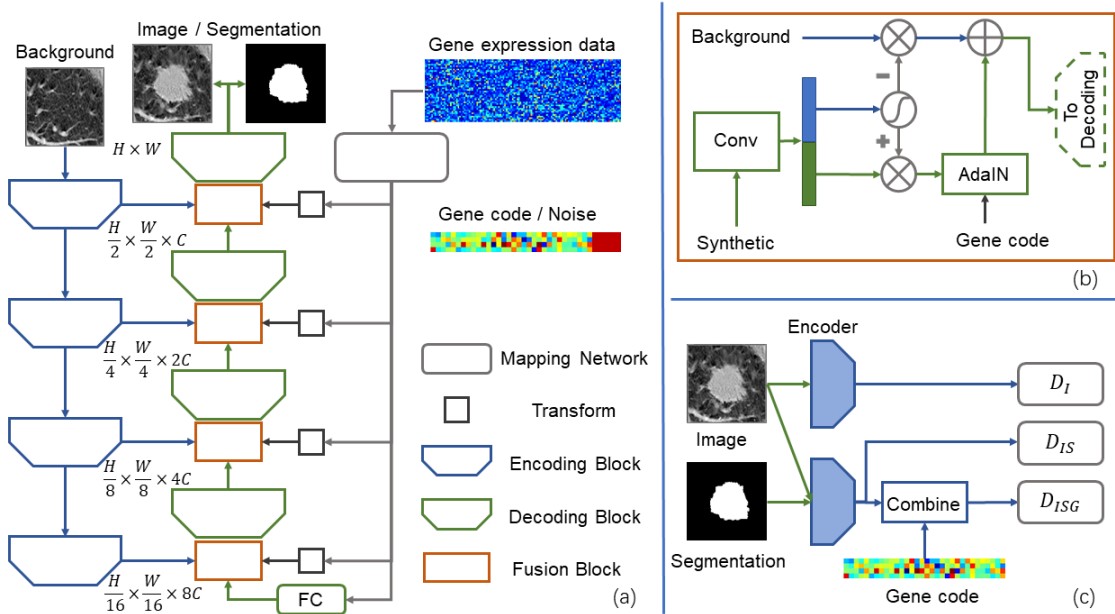

Figure 1: Proposed multi-conditional GAN for radiogenomic map learning and nodule synthesis. (a) Generator utilizes both background image and gene code to synthesize image together with nodule segmentation. (b) Fusion block at each resolution layer helps to fuse the information from background with that from previous layer and gene code. (c) With image, segmentation, and gene code, discriminator distinguishes three types of real/fake scenarios.

## 2.1. GAN architecture

Fig.1 (a) illustrates the structure of the proposed generator. From background image and gene expression data, it generates a synthetic image with a nodule characterized by the genomic data, and

situated within the background image. Meanwhile, it also produces a binary segmentation mask of the generated nodule. Structure-wise, it consists of three parts: encoding of the background image (left), encoding of gene expression data (right), and information fusion for synthetic image/mask generation (center). The proposed GAN handles two major challenges: object/background separation and blending, and image/gene representation fusion.

For the first challenge, unlike previous inpainting-based methods, the proposed network does not remove any portion of the background image. Instead, it models object/background within the network via two strategies: 1) a fusion block at each resolution level to control the overlapping between generated object with reference background, and 2) an auxiliary output of segmentation mask to guild such separation. At each stage, it performs a "soft" blending of object/background information, and therefore ensures spatial continuity of the outcome synthetic image. Through such mechanism, the gene code controls both the foreground and the meaningful region where lesion blends with the background, while separated from the irrelevant background region. The use of auxiliary output of segmentation is the same as baseline method (Park et al., 2018), it is a part of the synthesis process with the aim of ensuring a good transition/fusion between object and background.

For the second challenge, existing work in computer vision (Park et al., 2018) used word embedding to produce a base image that can be combined with background at the bottleneck layer of a encoder-decoder network. One major difference between word embedding and gene representation is that word is much more closely related to image. Therefore, we choose to model the gene information as the abstract "style" of an image, and use style transfer techniques to guide the synthesis process (Karras et al., 2018). Specifically, high-dimensional gene expression data is encoded with a mapping network, this can be a few fully-connected (FC) layers (Karras et al., 2018), or more sophisticated conditioning augmentation block (Zhang et al., 2017)(Park et al., 2018). Here, for better interpretability of gene encoding, we choose to use two FC layers to encode the raw gene data $g$ to a vector $\phi(g)$. $\phi(g)$ is further concatenated with a noise vector $n$ to be used as the base style map. Meanwhile, the background image is encoded with conventional image encoders consists of convolutional layers (Park et al., 2018). With image features and gene map, we use a series of fusion blocks to combine the information. The fusion block take image features from both background and previous step, together with gene "style" map to achieve: 1) proper separation of object/background, and 2) proper fusion of image/gene information.

### 2.2. Object separation and information fusion

Fig.1 (b) described the proposed fusion block. At each resolution level, we have three information feed-ins: background image feature, gene map, and synthesis image feature from the previous layer. Since it contains information for both object and background, the synthetic features are first further encoded via two layers of convolution and batch normalization. During this process, the channel number is doubled. The resulting code is then split into two parts: the first half used as a weight map to control how much object/background information will be passed to further processing at this layer; while the other half will be used as object feature map.

As shown in the figure, both object/background feature maps will be controlled by element-wise multiplication with the weight map ($+$) and its inverse ($-$). Map $+$ suppressed the background information, passing mainly nodule features to be normalized by gene code. This is because we need to formulate the system so that gene code has less to do with background and more to control nodule appearance. Meanwhile, map $-$ suppressed the information where nodule will be generated,

reinforcing background information to align with input image. This process is similar to the hard cropping operation for inpainting, but in a soft and progressive manner within each layer of network. Gene code then controls the "style" of the synthetic nodule via an adaptive instance normalization (AdaIN) layer (Karras et al., 2018). Finally, the two are added together and fed to upsampling/decoding layer.

In this way, we achieved "soft" separation of lesion region and background, where genomic and imaging information are fused together. Comparing with completely erasing part of the image as "inpainting", the weight map is a learnt probability, which retains the information necessary for modelling the interaction between object and background. Comparing with word embedding synthesis (Park et al., 2018), our strategy strengthened the object/background separation because the gene map is to be applied mostly to the object region and has little impact over background.

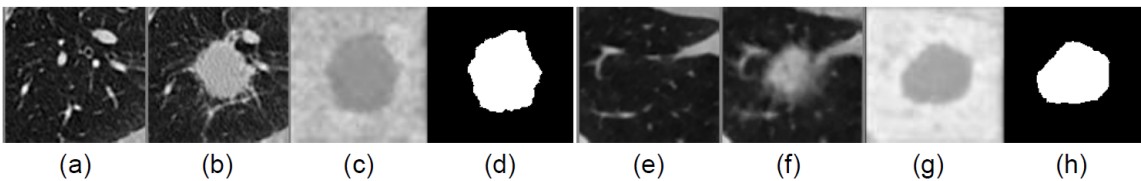

| (a) | (b) | (c) | (d) | (e) | (f) | (g) | (h) |

Figure 2: Examples of proposed synthesis GAN: (a, e) background image, (b, f) synthesized nodule image, (c, g) background weight image, (d, h) segmentation mask.

Fig.2 shows two examples (a-d) and (e-h) for the proposed GAN. (a) is the background image, (b) is the synthesis result, (c) is the background weight map, and (d) is the resulting mask. (e-h) is the same as (a-d) but for another case with ground-glass opacity. It can be observed that the original background image does not get change significantly with major structures preserved. At the same time, the synthesized nodule fused naturally with background image, this is especially important for a ground-glass case. It can also be seen that for two background images under different reconstruction, (e) being smoother than (a), the sharpness of the resulting nodule image also aligns well with the background image. Another important observation is that the attention map and mask coverage is larger than the solid lesion region, which confirms our hypothesis that both nodule and its interaction with background tissue, i.e. the transition region in between, are modeled under control of gene code.

### 2.3. Training strategy

The proposed GAN encodes genomic features as a vector, and outputs both image and segmentation. Here, the discriminator is illustrated in Fig.1 (c), following the method in (Park et al., 2018). The input to the discriminator is a tuple of image-segmentation-gene code. Two encoders are utilized to encode: 1) image for discriminator $D_I$, and 2) image-segmentation pairs for discriminator $D_{IS}$. The second encoder's output is further combined with gene code $\phi(g)$ and further encoded via convolution, batch normalization, and leaky ReLU activation layers for discriminator $D_{ISG}$. Discriminators are trained with least squares loss functions (Mao et al., 2017). Given image $x$, matched gene code $g$, and matched segmentation mask $m$, tuples to be discriminated against it include cases containing mismatched gene code $\bar{g}$, mismatched segmentation mask $\bar{m}$, synthetic image $G_x$, and synthetic mask $G_m$. Let $p_d$ and $p_G$ denote the distributions of real and synthetic data, we have

$x, g, m, \bar{g}, \bar{m} \sim p_d$ and $G_x, G_m \sim p_G$. With different combinations, we have

$$L_{D_I} = \mathbb{E}[(D_I(x) - 1)^2] + \mathbb{E}[D_I(G_x)^2]$$
$$L_{D_{IS}} = \mathbb{E}[(D_{IS}(x, m) - 1)^2] + \mathbb{E}[D_{IS}(x, \bar{m})^2] + \mathbb{E}[D_{IS}(G_x, G_m)^2]$$
$$L_{D_{ISG}} = \mathbb{E}[(D_{ISG}(x, m, g) - 1)^2] + \mathbb{E}[D_{ISG}(x, \bar{m}, g)^2]$$
$$+ \mathbb{E}[D_{ISG}(x, m, \bar{g})^2] + \mathbb{E}[D_{ISG}(G_x, G_m, g)^2]$$

For training the generator, the background reconstruction loss is added to guide the feature extraction of background image during synthesis. Let $G_{\bar{M}}$ be a morphological eroded version of segmentation mask $G_m$'s inverse (i.e. background region), $\odot$ denote element-wise multiplication, the $L1$ loss is computed over background between synthetic image $G_x$ and base image $x$:

$$L_G = \mathbb{E}[(D_I(G_x) - 1)^2] + \mathbb{E}[(D_{IS}(G_x, G_m) - 1)^2]$$
$$+ \mathbb{E}[(D_{ISG}(G_x, G_m, g) - 1)^2] + \lambda\mathbb{E}[\|G_x \odot G_{\bar{M}} - x \odot G_{\bar{M}}\|_1]$$

## 3. Experiments and Results

We evaluate the proposed method using the publicly available NSCLC dataset (Bakr et al., 2017). This radiogenomic dataset is built upon a NSCLC cohort of 211 subjects. Together with CT images and segmentation maps of the tumors, a subset of 130 subjects also have RNA sequencing data from surgically excised tumor tissue samples. After removing all gene with NaN values, we end up with a 5172-dimensional gene vector for each case. A $60 \times 60 \times 60$ mm$^3$ volume-of-interest (VOI) centered at each nodule is first cropped from the original image. Then 2D slices with nodule presence are extracted as training samples. In total we have 3736 training image slices.

Background images are created as following: lung region is first segmented for each image, then the nodule regions are excluded from lung mask. Next, distance transform is computed for the resulting mask, and centers are selected at a random location 5 to 25 mm from the mask boundary. Around each center, a $60 \times 60 \times 60$ mm$^3$ VOI is cropped and 20 random slices are extracted from each VOI.

Our proposed method has two outputs: 1) realistic and controlled generation of nodule images, and 2) radiogemonic map learning that links gene information to image features. Since there is no prior work achieving these goals, we compare the proposed method against an in-painting method (Jin et al., 2018), and a baseline method (Park et al., 2018) (2D multi-conditional natural image synthesis).

Fig. 3 shows the performance of image synthesis with multi-conditional GAN. First row include 7 training images. The algorithm uses the gene information from each of them, together with background images from second row, to synthesize image with nodules. Third row is the results from in-painting method, fourth row from baseline method, and the last row is from the proposed method. As shown in the image, in-painting method does not have control over the appearance of the nodule, and suffers from the discontinuity along the cropping boundary. Although more sophisticated in-painting methods can potentially reduce this missing information issue; by keeping all information from the background image rather than cropping out part of it, the network essentially has more information to prevent discontinuity along the cropping boundary from happening. As compared with baseline, the proposed method generates more images that are less blurred/corrupted (e.g. last column). Also, The resulting synthetic images have similar features as the reference training samples.

On the other hand, we do notice that the current model has a certain degree of mode collapsing issue. While arguably, the proposed method still has an edge on the relative association to the reference images. For example, Column 5 and 7, where 5 has much fuzzier boundaries than 7, the proposed catches this feature (in a relative manner), but the baseline method generates a flipped version.

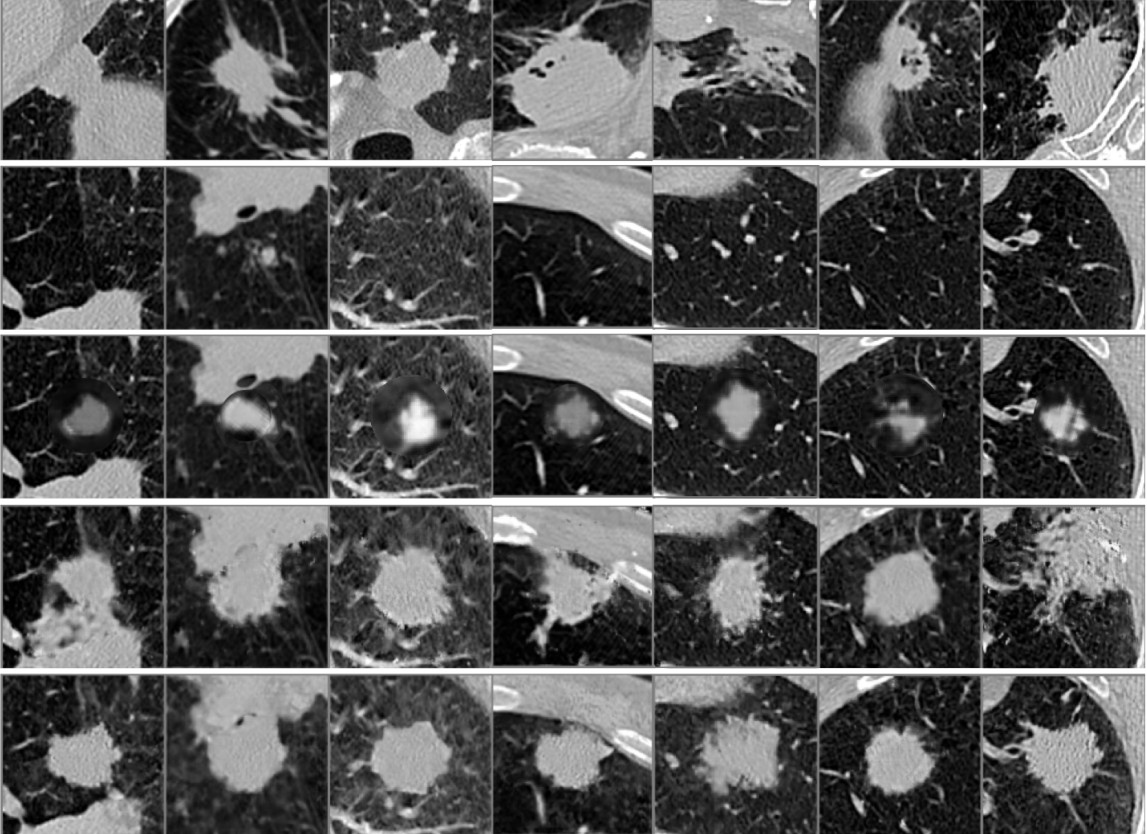

Figure 3: Result of nodule synthesis, first row: training image, whose genomic information is used to synthesize each column; second row: background image; third row: synthetic image by in-painting method (Jin et al., 2018); fourth row: synthetic image by baseline method (Park et al., 2018); last tow: synthetic image by the proposed method.

Table 1: Quantitative evaluation of image synthesis quality

| Method | MSE | SSIM | PSNR |
|---|---|---|---|
| In-painting | 0.0121 | 0.76 | 20.94 |
| Baseline | 0.0065 | 0.74 | 22.05 |
| Proposed | 0.0044 | 0.88 | 24.52 |

To further evaluate the results quantitatively, we used common image quality measurements including MSE, SSIM and PSNR within specifc regions between generated image and background

image. Results are listed in Table. 1. As shown in the table, the proposed have better performance over the two methods under comparison. The numbers, though, should only be used as a reference and interpreted with caution. This is because inherently, the in-painting method and the proposed method follows different logic. The in-painting method removes center sphere of the original image, so there is no change outside the sphere, and the major issue is the discontinuity along the sphere boundary. Here we calculated the measurements of in-painting within 1.5×radius sphere (otherwise any numbers can be achieved just by adjusting the sphere). The baseline and proposed method, on the other hand, try to synthesize nodule together with its fusion feature with the background, so the entire image got changed during synthesis. The numbers are calculated as an overall assessment, and may not be able to capture different aspects of each method. Therefore, we recommend readers to interpret them carefully, and refer to the qualitative results.

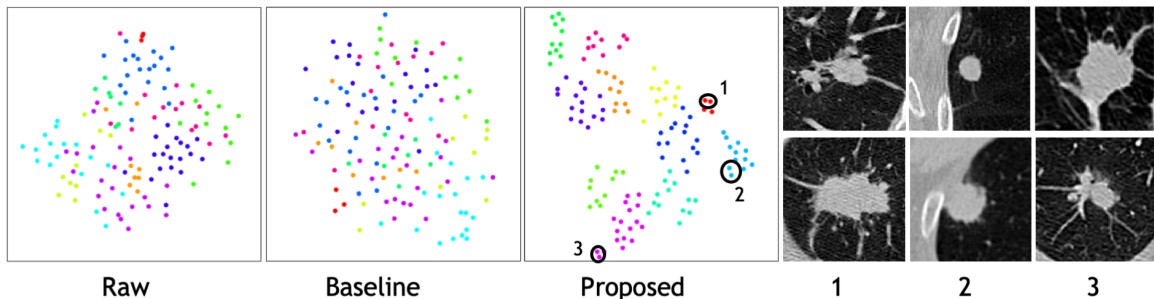

Figure 4: Distribution of gene coding illustrated by 2D t-SNE map (van der Maaten and Hinton, 2008) : raw gene (5172-D) and gene code produced by baseline method (128-D) does not show obvious separation, while gene code produced by the proposed method (128-D) showed feasibility for clustering. Three groups of samples are drawn from clusters formed according to distance, and their corresponding image are shown.

To check how well genomic information and image feature can be related with the proposed method, we showed the the radiogenomic correlation of the trained network in Fig. 4. In this experiment, the synthetic images are not involved, we look at the original image and comparing their raw / learnt gene codes. Here, we would like to see whether the similarities (from a distribution point of view) in gene codes can reflect their similarities in the image appearance. Supposedly, the closer gene codes generated by the proposed method can - to a certain degree - be related to similar image appearance.

Therefore, we project both the raw and the learnt gene codes using t-SNE (van der Maaten and Hinton, 2008) to generate a 2D representation of genomic data for easier visualization. The color groups in Fig. 4 are formed by the rough clusters from the proposed coding method, and we keep the same color based on this map for each individual point cross all three maps. As shown, no clusters can be reasonably formed from the raw gene vector, or the one produced by baseline method. This shows that the cluster-separability of the original gene code is quite weak, and hence confirms the difficulty of genome clustering if it is treated independently from image, as used in almost all previous radiogenomic studies. Since genes are clustered with such high uncertainties, this could potentially explain why the final findings are mostly weak in previous studies. With the proposed method, the resulting gene code can be separated to clusters. By examining their corresponding

image, we can observe some general correlation between gene cluster and the image features such as nodule shape and boundary smoothness. Our approach opens a new perspective beyond common practice and alleviates this issue as shown by the t-SNE map. The end-to-end framework removes the need for handcrafted features for finding relations between the imaging and genetic data.

## 4. Discussion and Conclusion

In this work, we use a multi-conditional GAN, coupled with a new structure of style control and fusion, to effectively generate realistic nodules whose appearance is controlled by its genomic features. Without erasing any portion of condition image, our method is superior over state-of-the-art method in object realism and object/background separation and fusion. An end-to-end mechanism is achieved to holistically model and correlate various features. As such, our approach can provide not only an effective and controllable means to generate diverse nodules, but also a discriminative radiogemonic map linking genomic and image features. To our best knowledge, this is the first attempt to address radiogenomic mapping by fusing the image and gene information together within an end-to-end GAN network. Currently, this work is proof-of-concept given that the data size is limited and there are still many unanswered questions. Conducting a study to make clinically convincing would be a long journey. There has been some clinical research that studies the characteristics of lung nodules and their correlations to associated gene coding. It is rather a sophisticated and ongoing research topic. With this paper, we would like to inspire the community about what we can do with gene codes and imaging data using deep learning. We show promising results and hope others may follow exploring what can be done in finding the correlation, which is less explored but can potentially have clinical impact.

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
