# OpenReview forum: "Correlation via Synthesis: End-to-end Image Generation and Radiogenomic Learning Based on Generative Adversarial Network"
_MIDL.io/2020/Conference — MIDL 2020_

### Official Review · AnonReviewer3 · 2020-03-11
**Interesting perspective and results**

**Rating:** 3
**Confidence:** 4
**Recommendation:** Poster

**Summary:**

The proposed aims to explore the correlation between genetic codes and medical images by synthesising images using conditional-GAN. The multi-label conditional GAN takes in encoded gene information as the style to generate pathological regions, such as nodules, on given background images. The conditional GAN follows a similar structure as style-based GAN by Karras et a and MC-GAN by Park et all.


**Strengths:**

The paper combines the knowledge from two domains, i.e. the image and genetic information, from an interesting perspective.
The network architecture, losses and training details of the proposed method are clearly written.
The method outperforms the in-painting method and the baseline method, namely multi-conditional GAN by Park et al, in terms of several evaluation metrics. it also shows better clustering capability of gene codes than the other two methods.


**Weaknesses:**

The motivation for the work is not very strong.
The authors pose the problem: metagene clustering and image feature extraction are done as two separate tasks without considering their correlation. To improve this, one would like to have more complicated image feature representation, rather than hand-crafted ones, and take advantage of the correlation between genetic data and image features. However. there seems to be a missing link between the proposed method and the motivation in the introduction, as the proposed method does not directly answer the posed question, except for the clustering analysis.
Maybe the authors would like to discuss how the conditional generative model can be used for better feature representation or exploration of correlation.  Also it might be interesting to observe, for a given segmentation, what information from the gene expression data is encoded into the gene code.

**Justification Of Rating:**

The paper overall is well-structured and thoroughly evaluated. The perspective is also very interesting. However, it may require a smoother connection between the motivation and the proposed, and provide more analysis for the clustering part.

**Paper Type:**

validation/application paper

**Questions To Address In The Rebuttal:**

Some clarification is needed:

1) if it's understood correctly, the multi-conditional GAN generates both images and their segmentation, is there any mechanism to ensure that the generated images do produce those segmentations, for example, feed them into a segmentation network and compare the output with the generated segmentation?

2) Figure 4 shows the clustering using the baseline method and the proposed method, clusters are shown in different colours, what do these colours correspond to?

Minor issue:
Page 2 Line 3, To out best knowledge -> To our best knowledge

**Special Issue:**

no

---

> ### Author Response · Authors · 2020-03-27
> **Reply to reviewer's comments**
>
> Dear reviewer, thank you very much for your time, encouraging comments, and valuable opinions on this manuscript.
> We would like to clarify a few points regarding your questions:
>
> 1. Motivation: Thank you for the valuable suggestions. Indeed, the current study does not cover the comparison with metagene-radiomics correlation method. Thank for pointing this out. We will reformulate our approach as an alternative, rather than “improvement” without further evidence. Also, the direction of feature representation analysis is for sure the way to go in future study.
>
> 2. Mask: Yes, the segmentation mask is a part of the synthesis process, the aim of using it is to ensure a good transition/fusion between object and background. Since it is used by the baseline method (Park, et al). We did not perform further evaluation on this. We will further clarify this point. Thanks for the suggestion on testing the mask quality, this will definitely be helpful – we currently examine them visually, if the masks are sufficiently accurate quantitatively, they may be used for future training purpose.
>
> 3. Clustering: Each point corresponds to an image/gene pair in the study. The color means the rough clusters formed by the proposed method, and we keep the same color cross the three maps for each point based on this map. Note that in this experiment, the synthetic images are not involved, it is only for original image and their learnt gene codes. Here, we would like to see whether the similarities (from a distribution point of view) in gene codes can reflect their similarities in the image appearance. Supposedly, the closer gene codes generated by the proposed method can to a certain degree be related to similar image appearance.

---

### Official Review · AnonReviewer4 · 2020-03-14
**Radiogenomic based approach by using Generative Adversarial Networks**

**Rating:** 3
**Confidence:** 5
**Recommendation:** Oral

**Summary:**

The authors have developed a multi-conditional generative adversarial network (GAN) conditioned on both background images and gene expression code, to synthesize a corresponding image, in a dataset of NSCLC from TCIA. This paper talks about a new multi-modal data integration in the medical imaging research, using GANs. The paper has great potential, but leaves the reader hanging with no proper conclusion. A few additional application based sub-experiments can make this paper really impactful.

**Strengths:**

* Key Idea - GANs can be used in a multimodal integrated approach for radiogenomic studies
* A very impressive idea with tremendous potential
* Well written paper that clearly explains the methodology

**Weaknesses:**

* As a reviewer, I would have liked to see some validation. For example, once this radiogenomic integration method was established, the authors should have done a more deep dive analysis of how the synthetic nodule correlates with genomic information and the features of the original nodule. This was lightly touched by the tSNE, but this needs to be fleshed out more.
* Another potential approach - Use the TCIA cases.. they have multiple radiology and genomic datasets. Test your methods and its correlative analysis across various cancer sites.

**Justification Of Rating:**

A good methods based paper. The paper talks about the ways of how genomics and radiology data can be integrated. This is very relevant and aligned with the MIDL workshop and its vision. The paper has also been written  well, with informative figures.

**Paper Type:**

methodological development

**Special Issue:**

yes

---

> ### Author Response · Authors · 2020-03-27
> **Reply to reviewer's comments**
>
> Dear reviewer, thank you very much for your time, encouraging comments, and the fantastic recommendation! We definitely align with you for the future work on this project.  We in fact have concerns over the limited amount of data used in this current study, and will check TCIA for more images.

---

### Official Review · AnonReviewer2 · 2020-03-14
**Interesting work on radiogenomic learning with good results**

**Rating:** 3
**Confidence:** 5
**Recommendation:** Oral

**Summary:**

The paper presents a method to combine gene code with image features so as to generate synthetic images. The proposed network takes background image and gene expression and generates an image of lung nodule which is characterized by the genomic data. Along with the lung nodule image which is located within the background image, the network also generates a segmentation mask. Experiments are performed on NSCLC dataset.

**Strengths:**

1. The idea of generating images of lung nodules characterized by the gene code is quite interesting.
2. The overall approach can be considered as an original work.
3. The paper includes both qualitative and quantitative results.

**Weaknesses:**

1. It is mentioned in section 2, that inpainting leads to the loss of regional information and doesn't ensure spatial continuity. This might be not be completely true, as the goal of inpainting is to have realistic filing and a good inpainting network should take care of both regional information as well as spatial continuity.

2. In section 2, it is not clear, why there is a need to generate a segmentation mask and weight image.

3. In section 2.2 why map (-) suppresses the nodule information, as the background is already fed in as the input image.

4. Ablation study pertaining to the three discriminators is missing.

5. In figure 3, the fourth and fifth rows seem similar, so it would be difficult to visually evaluate them.

6. Overall, the clinical motivation to generate nodules images characterized by gene code is not convincing and may not be of great interest to the readers.

**Justification Of Rating:**

The paper tackles an interesting problem of generating nodule images conditioned on the gene expression data. The authors present extensive qualitative and quantitative results along with clustering results to show the radiogenomic correlation.

**Paper Type:**

methodological development

**Special Issue:**

yes

---

> ### Author Response · Authors · 2020-03-27
> **Reply to reviewer's comments**
>
> Dear reviewer, thank you very much for your time, encouraging comments, and valuable opinions on this manuscript.
> We would like to clarify a few points regarding your questions:
>
> 1. Inpainting: Indeed, we definitely agree with you on the goal of inpainting. Here, we try to point out that it may provide more information to the network if we keep all information from the background image rather than cropping out part of it, which may often lead to some discontinuity along the cropping boundary (Fig. 3).
>
> 2. Mask: Sorry for the confusion, the segmentation mask is a part of the synthesis process, the aim of using it is to ensure a good transition/fusion between object and background. Since it is used by the baseline method (Park, et al). We did not perform further evaluation on this. We will further clarify this.
>
> 3. Map (-): The purpose of map (-) is to softly and progressively “remove” the background information where the nodule is going to be painted. In fact, it is similar to the hard cropping operation for inpainting, but in a soft and progressive way within each layer of network.
>
> 4. Ablation for discriminators: Thanks for the comment, this method is based on baseline (Park, et al.) where the similar losses have been studied. To save space, we skip this part. We will clarify this point.
>
> 5. Synthesis: Thanks for the comment. The baseline (4th) appears to bring more unrealistic distortion to the background image. Revisiting these samples, we agree that we would definitely need a better balance between quality and variability when selecting the end model. As mentioned by another viewer, apparently, the current model has a certain degree of mode collapsing issue. While arguably, the proposed method still has an edge on the relative association to the reference images. For example, Column #5 and #7, where #5 has much fuzzier boundaries than #7, the proposed catches this feature (in a relative manner), but the baseline method generates a flipped version.
>
> 6. Motivation: Conducting a study to make clinically convincing would be a long journey. With this paper, we would like to inspire the community about what we can do with gene codes and imaging data using deep learning. We show promising results and hope many others will follow exploring what can be done in finding the correlation, which is less explored but can be potentially impactful, clinically.

---

### Official Review · AnonReviewer5 · 2020-03-19
**Very interesting approach with potential broad applications, but some interpretations need more evidence**

**Rating:** 3
**Confidence:** 4

**Summary:**

The paper presents an approach to tackle genetic and radiology images fusion via image painting of nodules in non-neoplastic lung images. An end-to-end deep learning GAN is presented with a bicephalic structure, processing simultaneously a gene map and a background image, and outputting a generated image and an associated predicted tumor segmentation mask. The system is trained using 3 discriminators, intended to discern generated images, segmentation maps mismatches and gene map mismatches. Experiments are conducted on a public dataset involving a total of 130 subjects. Most results are qualitatively reported.

**Strengths:**

The paper is well written: ideas are clear, language is formal, and most of the expected related works and introductory materials are present. There is substantial effort in benchmarking with appropriate approaches, although the task is relatively novel. Experiments on public data should be reproducible with the details provided by the authors.

**Weaknesses:**

There is not enough information about the metrics extracted. How are MSE, SSIM and PSNR computed ? between generated image and background image ?

Although this is arguable, results in Figure 3 indicate that there is less variability of shapes of proposed generated images (last row), than in Park and al. and less than the images associated to the input gene codes. Apart from this aspect, there is no doubt that the proposed yield the best generation quality. However, this could contradict one of the 2 claims of the authors, which is "that a discriminative radiogemonic map can be learnt via this synthesis strategy".
The gene coding results section does not tackle this aspect since it is only related to genomic information (taken from vectors outputted by the transformer). T-sne is also only a projection of the data onto a subdimensional space and is therefore arguably a good representation of the true data distribution. Besides, what is the meaning of colors in t-sne results? Are those training samples for both 3 methods?

End of 2.2 is in methods and should probably be in results.

What is the impact of the segmentation aspect of the approach? No benchmark is performed without segmentation, and no segmentation performance is reported. Overall, readers could benefit from more information about the loss behavior during training to better understand each discriminator's impact.

For broader audience target, authors should specify how to use the network to extract fused radio-genomics features maps for other tasks, as it is one of the 2 objectives.

**Detailed Comments:**

Part 3: " Also, The resulting" -> "Also, the resulting"

More implementations details could be added for further reproducibility (learning rate, optimizer, batch size, GAN tricks etc)

**Justification Of Rating:**

There is a lack of evidence in the second claim of the paper about fused representation, reducing score to weak accept.
Strong accept if authors show further evidence of fused representation of both genetic and imaging data.

**Paper Type:**

both

**Questions To Address In The Rebuttal:**

The image generation performance is clear. There remains some doubt on the propensity of the network to fuse genetics data with imaging ones. T-sne qualitative performance assessment does not take into account any image features. Can authors propose further evidence that genetic information is indeed integrated in the produced images, which would indeed illustrate the proposed framework can link genomic and radiomic data without dramatic loss of information?

**Special Issue:**

no

---

> ### Author Response · Authors · 2020-03-27
> **Reply to reviewer's comments**
>
> Dear reviewer, thank you very much for your time, encouraging comments, and valuable opinions on this manuscript.
> We would like to clarify a few points regarding your questions:
>
> 1. Metric: The metrics are computed between generated image and background image. However, to make the comparison as fair as possible, we restricted the computation of inpainting method within an extended cropped region. Such metric is still not ideal, so we added the comments that they should be interpreted with caution. We will further clarify this.
>
> 2. Synthesis variability: Thank you for the great point. Revisiting these samples, we agree that we would definitely need a better balance between quality and variability when selecting the end model. Apparently, the current model has a certain degree of mode collapsing issue. While arguably, the proposed method still has an edge on the relative association to the reference images. For example, Column #5 and #7, where #5 has much fuzzier boundaries than #7, the proposed catches this feature (in a relative manner), but the baseline method generates a flipped version. Again, we definitely need to better train the model to cover more variability following your suggestion.
>
> 3. Gene coding: Thank you for the question. This part definitely needs further clarification. In this part, the synthetic images are not involved, it is only for original image and their learnt gene codes. Here, we would like to see whether the similarities (from a distribution point of view) in gene codes can reflect their similarities in the image appearance. Supposedly, the closer gene codes generated by the proposed method can to a certain degree be related to similar image appearance.
> Therefore, we project the learnt gene codes to t-SNE for easier visualization, here the color means the rough clusters formed by the proposed method (and we keep the same color based on this map for each individual point cross the three maps). Then we took points that are close to each other and identify their corresponding original images, to see if the images are also similar, which is shown in 1/2/3.
> The logic is a bit different from the previous experiment, so please let us know if this clarifies your concern.
>
> 4. Segmentation: The segmentation mask is a part of the synthesis process, the aim of using it is to ensure a good transition/fusion between object and background. Since it is used by the baseline method (Park, et al). We did not perform further evaluation on this. We will further clarify this.
>
> 5. Radio-genomics features: Thank you for the suggestion. First, we would like to further find the connections between gene and image; then we may be able to use them together (simple concatenation or more sophisticated fusion) for other tasks. We will add these to the discussion part.
>
> 6. Integrating genomic information: There has been some clinical research that studies the characteristics of lung nodules and their correlations to associated gene coding. It is rather a sophisticated and ongoing research topic and our motivation here is to investigate the connection using a machine-learning-based method. We will further corroborate our findings with clinical research outputs in the future.

---

### Meta-Review · Area_Chair1 · 2020-04-06
**MetaReview of Paper58 by AreaChair1**

**Rating:** 3
**Recommendation For Accepted Papers:** Poster

**Metareview:**

All the reviewers seem to agree that the ideas presented in the paper and the combination of image and genetic information are very interesting while the paper is clearly presented and written. I agree that the idea is very interesting and it can be also very interesting for the community. The authors addressed the comments of the reviewers properly and I encourage them to incorporate them in the final version of the paper to improve their final version.

**Paper Type:**

methodological development

**Special Issue:**

yes

---

### Decision · Program_Chairs · 2020-04-11

Accept